# Daily Simulation of the Rainfall–Runoff Relationship in the Sirba River Basin in West Africa: Insights from the HEC-HMS Model

Idi Souley Tangam [1,2], Roland Yonaba [1,*], Dial Niang [1], Mahaman Moustapha Adamou [2], Amadou Keïta [1] and Harouna Karambiri [1]

1 Laboratoire Eaux, Hydro-Systèmes et Agriculture (LEHSA), Institut International d'Ingénierie de l'Eau et de l'Environnement (2iE), Ouagadougou 01 BP 594, Burkina Faso; idi.souley@2ie-edu.org (I.S.T.); dial.niang@2ie-edu.org (D.N.); amadou.keita@2ie-edu.org (A.K.); harouna.karambiri@2ie-edu.org (H.K.)

2 Faculté d'Agronomie, Université Abdou Moumouni de Niamey (UAM), Niamey BP 10662, Niger; moustapha_a@yahoo.com

* Correspondence: ousmane.yonaba@2ie-edu.org

**Abstract:** This study focuses on the Sirba River Basin (SRB), a transboundary West African catchment of 38,950 km$^2$ shared by Burkina Faso and Niger, which contributes to flooding downstream in Niamey (Niger). The study uses the HEC-HMS hydrological model to explore the dynamics of the daily rainfall–runoff relationship over the period 2006–2020. The model is calibrated using observed rainfall at 13 meteorological stations within the river basin and observed discharges at the Garbey Kourou hydrometric station outlet. Two types of simulation are compared: (i) a continuous simulation (CS) over the period 2006–2020 and (ii) an event-based simulation (ES) using selected major flood events in 2010, 2012, 2013, 2015 and 2020. The results showed satisfactory model performance under both modeling schemes ($R^2$ = 0.84–0.87 for CS and $R^2$ = 0.94–0.98 for ES), with a superior performance of ES over CS. Also, significant differences in the distribution of calibrated model parameters for the percent impervious and the attenuation flood wave factor were observed. A sensitivity analysis revealed that the curve number, initial abstraction, lag time and routing time factors were influential on the model outputs. The study therefore underscores the model's robustness and contributes crucial insights for flood control management and infrastructure planning in the SRB.

**Keywords:** flood; HEC-HMS; hydrological modeling; Sirba River Basin; surface runoff





## 1. Introduction

The hydrological regimes of many rivers in the world have undergone major disruption in recent decades, mainly as a result of the combination of climatic variability and anthropogenic factors [1–3]. This climatic variability is reflected in alternating dry and wet cycles [4], with very pronounced extremes, which have led to a significant increase in flooding worldwide. Over the last twenty years, floods have been acknowledged as the most frequent and devastating natural disaster, affecting around 1.65 billion people and causing damage estimated at USD 651 billion worldwide [5,6].

In West Africa, Niger, like many other African countries, is confronted with these flooding events, particularly in Niamey, the capital city, which has experienced several cycles of large-scale flooding [7–10]. Currently, both fluvial floods, caused by overbank flow of the Niger River, and pluvial floods constitute a threat to the city [11]. Flooding occurs due to various factors such as heavy rainfall, full overtopping flows in the Niger River and climate vulnerability compounded by the region's dependence on rainfed agriculture [12], rapid population growth and recurring challenges linked to poverty and political instability [13]. The causes of floods in the region are difficult to determine, and the physical characteristics of catchments such as geomorphology and land cover, although

poorly documented, are thought to play an important role in the surface runoff generation processes [14–17]. The occurrence of floods has increased in recent years, with about 8–12 events per year compared to only 2 events in the past [9,11]. The 2010 flood, for example, was caused by heavy rainfall, which resulted in the highest water levels in the Niger River in 80 years [18]. Additionally, the region's vulnerability to flooding is further exacerbated by water scarcity, longer dry seasons and the impacts of higher temperatures, which may trigger new conflicts and forced migration. The spatiotemporal dynamics of suspended particulate matter in the Middle Niger River Basin is also likely to contribute to the understanding of flooding in the area. Furthermore, hydrographs of the Niger River at the Niamey hydrological station provide insights into the water contributions from the Sirba and Gorouol upper river basins, which can lead to flooding in the region [13].

Flood events cause significant losses and damages, including loss of life, displacement of people and destruction of homes and infrastructure [5,19,20]. An in-depth assessment of the flow process in the Niger River catchment, including the accurate estimation of flood peak and flood runoff volumes, is essential to contribute to flood management, planning and prevention. In this line of thought, according to previous studies, the Sirba River Basin (SRB) plays a major role in the flooding events occurring in Niamey, the capital city of Niger, with a contribution estimated at 80% [21]. Indeed, the SRB is the largest tributary of the Niger River, where a 30% increase in surface runoff has been observed since the 1970s [18,22–24].

Hydrological modeling offers an interesting framework for large-scale analysis and assessment of the rainfall–runoff relationship [25–29], especially in the context of the SRB. Several hydrological modeling studies have been conducted in the SRB in West Africa, focusing on various aspects such as land surface modeling [30], statistical discharge time series analysis [31–33], hydrological rainfall–runoff assessments [21,23,34–38] and the development of flood early warning systems [39]. Although these studies shed light on the understanding of hydroclimatic behavior and variability in the region, little work has been carried out on the possible contribution of the SRB to the onset of fluvial floods downstream of the SRB, in Niamey urban city. In this regard, the semi-distributed and physically based *Hydrologic Engineering Center–Hydrologic Modeling System* (HEC-HMS) model [40] has proven to be effective for long-term continuous and event-based surface runoff simulation, especially regarding peak flows and volumes during flooding events [10,41,42]. The model has never been applied in the region [10]; moreover, it can easily be coupled with the *HEC–River Analysis System* (HEC-RAS) for flow river hydraulics analyses and flood mapping analyses in urban environments [10,43,44].

The study objectives are twofold: (1) to analyze the rainfall–runoff relationship in the SRB through the application of the HEC-HMS hydrologic model and (2) to quantify the sensitivity of surface runoff volumes and peak flows to the HEC-HMS model parameters. The analysis period defined for this study is the period 2006–2020, for which hydrometeorological records are readily available for proper rainfall–runoff modeling and assessment. The results of this study will shed light on the rainfall–runoff relationship in the SRB and further help to assess how the catchment contributes to flooding downstream of the SRB boundaries.

## 2. Materials and Methods

### 2.1. Study Area Description

The study area is the Sirba River Basin (SRB), which is the main tributary of the Niger River in West Africa. It is a transboundary catchment shared between two neighboring countries, Burkina Faso and Niger (Figure 1). Almost 90% of the total catchment area is located in the northwest region of Burkina Faso, while 10% of the catchment area lies within Niger boundaries. The catchment drains a total area of 38,950 km$^2$ at Garbey Kourou hydrometric station and is located between the northern latitudes 14°23′30″ and 13°55′54″ and longitudes 1°27′00″ west to 1°23′4″ east. The flow regime in the catchment is intermittent and mainly driven by rainfall. The climate in the SRB is of Sahelian semi-arid

type, with an average annual rainfall ranging from 400 mm in the north to 800 mm in the south (over the period 1998–2020) [45]. The geological basement of the catchment consists of two major units, a granitic formation and the Birimian.

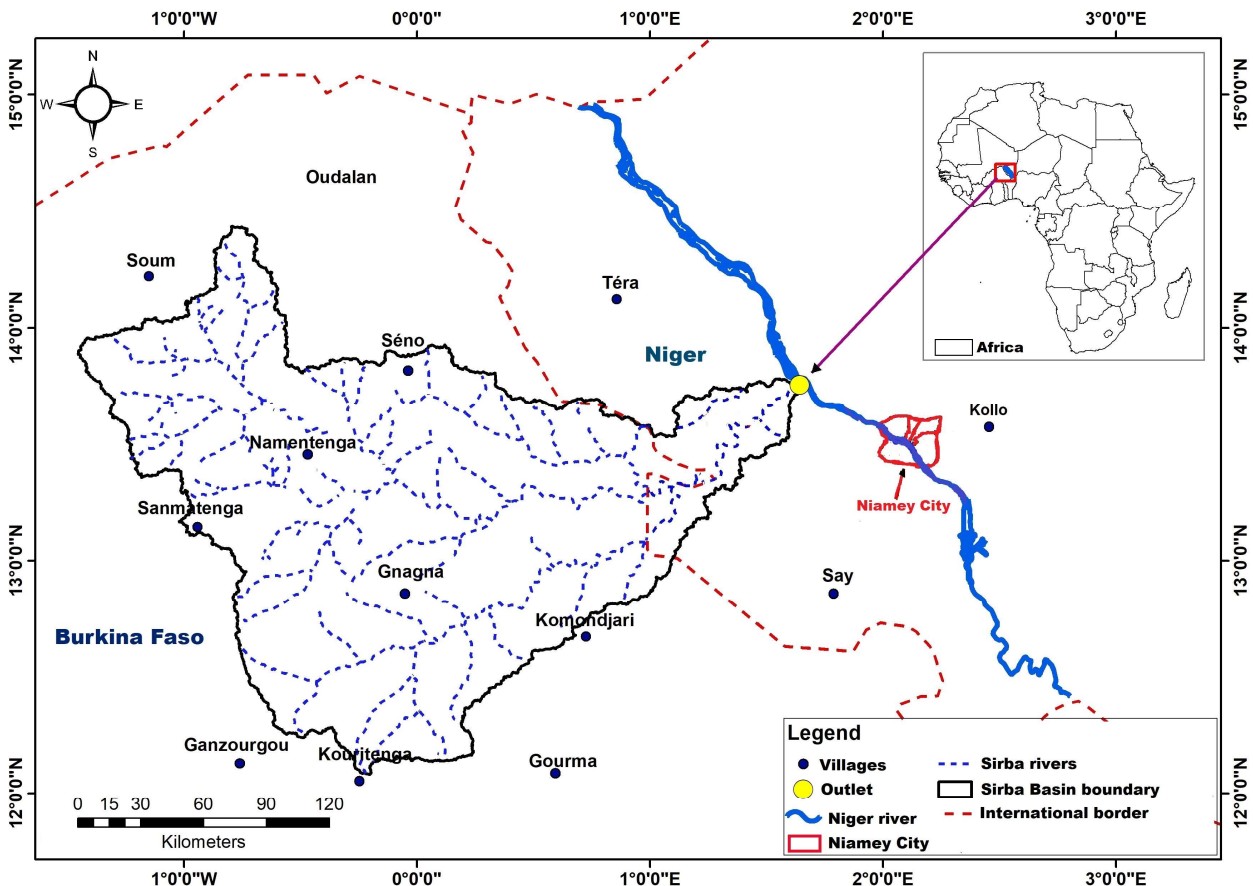

**Figure 1.** Location of the transboundary Sirba River Basin (SRB) in West Africa.

*2.2. Data Used in This Study and Preprocessing Steps*

In this study, daily rainfall data were collected from the National Meteorology Agency in Burkina Faso (ANAM-BF) and the National Meteorological Directorate of Niger (DMN-Niger). The data were collected at 13 meteorological stations, including three synoptic stations in Burkina Faso (Fada N'Gourma, Bogandé and Dori) for the period 2006–2020 (i.e., 15 years). The raw data included a few gaps (between 0.3% and 5.7%), which were further infilled using k-nearest neighbor and linear regression interpolation and methods [46,47]. The rainfall data were therefore spatially averaged using Thiessen's polygon method [48]. Figure 2 shows the location of the stations in the SRB and the spatial repartition of the annual rainfall in the watershed over the period 2006–2020, which varies between 400 mm and 800 mm, following a north-to-south gradient.

Elevation data (Figure 3a) were collected from the ALOS PALSAR Digital Elevation Model (DEM) of 12.5 m spatial resolution, which was accessed through the Alaska Satellite Facility (ASF) platform (https://search.asf.alaska.edu, accessed on 30 September 2023). The data provide radiometric high-resolution terrain-corrected elevation derived from L-band synthetic aperture radar (SAR) [49].

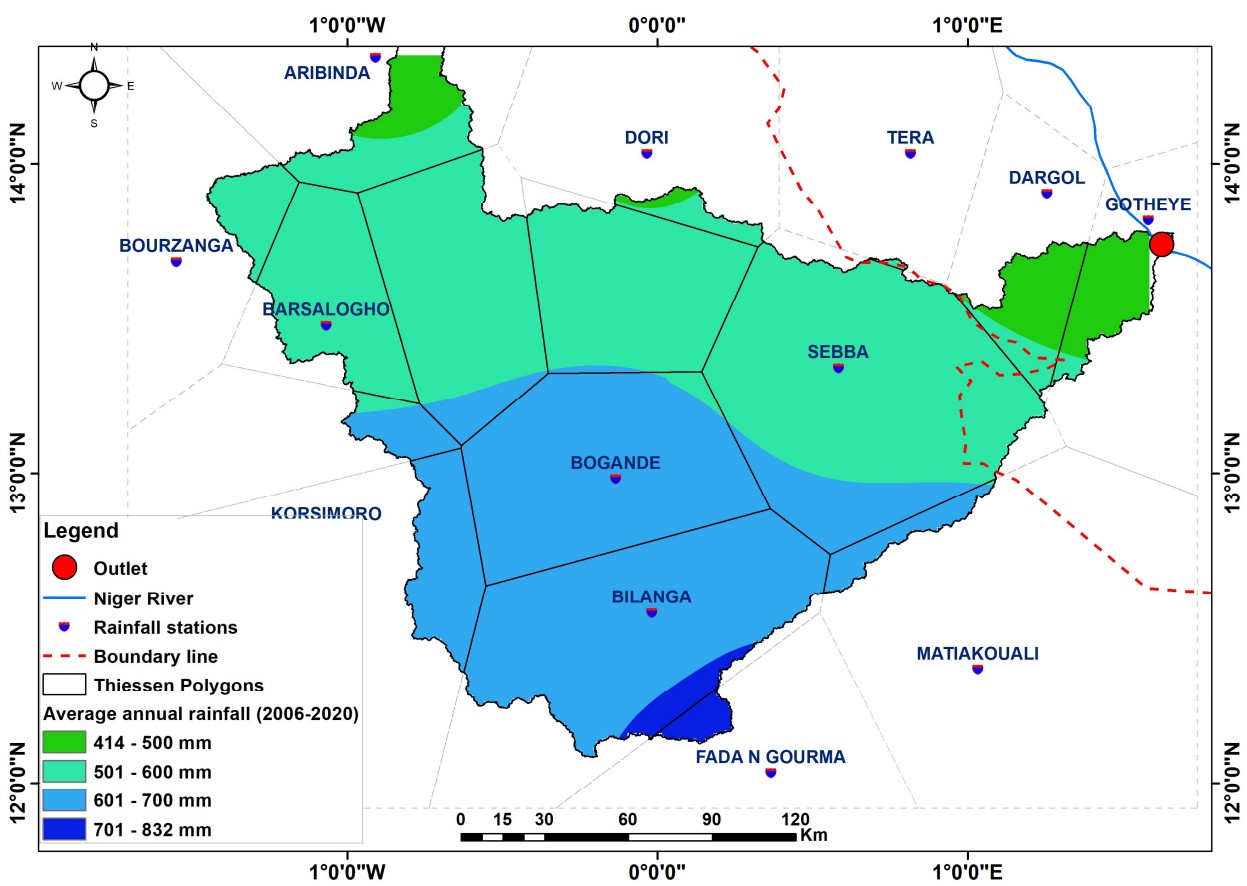

**Figure 2.** Location of rainfall stations and annual rainfall in the SRB over the period 2006–2020.

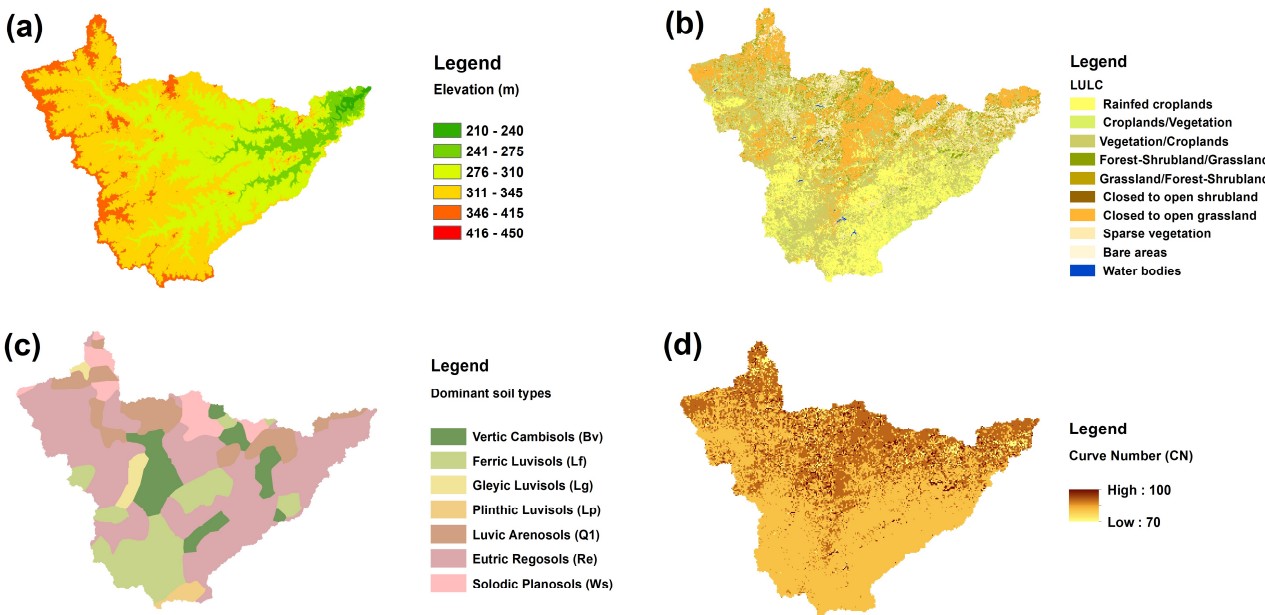

**Figure 3.** Physical description of the SRB: (**a**) elevation, (**b**) land use/land cover (LULC), (**c**) dominant soil types and (**d**) Curve Number (CN) grid maps.

Land use/land cover (LULC) data used in this study (Figure 3b) were derived from the *GlobCover 2009* global land cover map (http://due.esrin.esa.int/page_globcover.php, accessed on 30 September 2023) produced by the European Space Agency (ESA) initiative, which delivers global composites and land cover maps at 300 m spatial resolution from MERIS/ENVISAT sensors [50,51].

Soil data (Figure 3c) in the SRB catchment were collected from the Harmonized World Soil Database (HWSD, version 1.2), available from the Food and Agriculture Organization (FAO) website portal (https://www.fao.org/soils-portal/data-hub/soil-maps-and-databases/harmonized-world-soil-database-v12/, accessed on 30 September 2023) [52]. According to the data, the texture of 41% of the soils in the SRB are loamy sandy (Hydrologic Group C, moderate runoff potential), 45% are loamy clay sand (Hydrologic Groups C and D, moderate to high runoff potential), 7% are sandy (Hydrologic Group B, low to moderate runoff potential) and 7% are clayey (Hydrologic Group D, high runoff potential) [53]. The soil data were further processed through the ArcHydro tool within the ArcGIS software [54] to generate a Curve Number (CN) [53,55] grid map (Figure 3d).

The daily observed discharge values at the Garbey Kourou hydrometric station, which is the SRB outlet, were provided by the Niger River Basin Authority for the period 2006–2020. On average, the annual discharges show a high interannual variability, ranging between 313 $m^3/s$ (in 2014) to 1209 $m^3/s$ (in 2020), with an average value of 679.1 $\pm$ 284.2 $m^3/s$.

### 2.3. Rainfall–Runoff Hydrological Modeling

#### 2.3.1. HEC-HMS Model Presentation

The *Hydrologic Modeling System* (HEC-HMS) is a comprehensive software designed to simulate the complete hydrologic processes of dendritic watershed systems, used for various purposes such as designing and operating projects, regulating floodplain activities and conducting flood hazard mapping [40,56]. HEC-HMS incorporates numerous hydrologic analysis procedures, such as event infiltration, unit hydrographs and hydrologic routing, as well as methods for continuous simulation, including evapotranspiration and soil moisture accounting [57]. It also provides advanced capabilities for gridded runoff simulation, model optimization, forecasting streamflow, depth-area reduction, assessment of model uncertainty, erosion, sediment transport and water quality [40,58]. In this study, the HEC-HMS model is used to simulate daily runoff over the period 2006–2020 in the SRB catchment.

#### 2.3.2. Catchment Delineation and Model Preparation

The ALOS PALSAR DEM data were hydrologically processed within the ArcHydro tool [59] to delineate the SRB catchment and subcatchment boundaries along with their physical and morphometric characteristics such as average longitudinal slopes, area, flow path length, reach segment and centroid location. Following [60], to account for spatial variability in soil surface conditions in this study, a subdivision into homogeneous subcatchments was carried out. In total, 12 subcatchments, ranging from 1548.1 to 6992.4 $km^2$ in size, were defined, as presented in Figure 4.

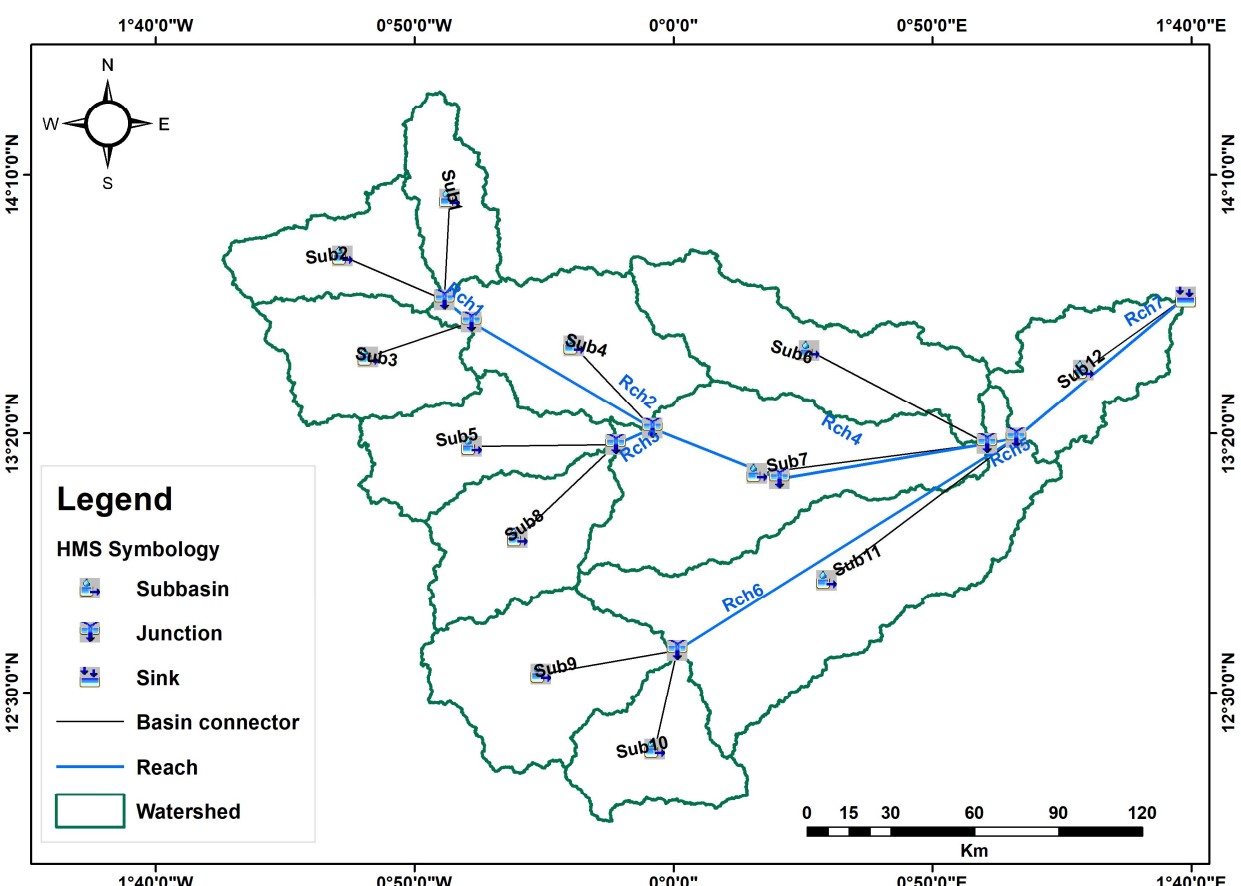

**Figure 4.** HEC-HMS model setup for hydrological modeling in the SRB.

### 2.3.3. Loss Model

The loss model calculates surface runoff volume from the supplied rainfall [40,58]. In this study, the Soil Conservation Service–Curve Number (SCS-CN) loss method [53,61] is applied for its simplicity and wide applications, which have proven to be effective in previous modeling attempts in similar environments [1,16,29,62]. Equation (1) describes the SCS-CN [53]:

$$Pe = (P - Ia)^2 / (P - Ia + S), \quad Ia = 0.2S, \quad S = 25400/CN - 254, \tag{1}$$

where $Pe$ is the excess rainfall (in mm), $P$ is the rainfall (in mm), $Ia$ is the initial abstraction (in mm) and $S$ is the potential maximum retention (in mm) and is inversely proportional to the Curve Number ($CN$), which is inferred from the soil type [53]. In the case a given sub-watershed $i$ has a different soil type, a weighted $CN$, noted $CNw_i$, is necessary and is computed from Equation (2):

$$CNw_i = \sum_{i=1}^{n} \frac{A_i \times CN_i}{A_t}, \tag{2}$$

where $A_i$ is a given sub-watershed $i$ area of Curve Number $CN_i$ and $A_t$ is the total subwatershed area.

### 2.3.4. Transform Model

The transform model converts excess rainfall into direct runoff [40,58]. In this study, the SCS Unit Hydrograph (SCS-UH) model is selected for this purpose. The model depends on a lag time parameter ($Tlag$), which is related to the concentration time ($Tc$). In this study, the initial values for $Tlag$ in each sub-watershed were calculated from Equation (3) [53,63]:

$$Tlag = 0.6Tc, \quad Tc = \frac{l^{0.8}(1000/CN - 10)}{1900S_0^{0.5}}, \tag{3}$$

where $Tlag$ is in minutes, $Tc$ is in hours, $l$ is the longest hydraulic path for a given sub-watershed (in m) and $S_0$ is the average slope gradient (in %).

### 2.3.5. Routing Model

In this study, the Muskingum method is used for routing model hydrographs at given outlet points [64]. This model uses two input parameters, which are the flood wave travel time ($k$) through a given routing reach and the attenuation flood wave factor ($x$), both computed through Equation (4):

$$S = k[xI + (1-x)Q], \tag{4}$$

where $I$ is the inflow, $Q$ is the outflow and $S$ is the storage. The weighting factor $x$ is bounded between 0 (maximum attenuation, mild slope channel and over-bank flow) and 0.5 (minimal attenuation, steeper channel and full wedge), while $k$ ranges between 1 and 150 h [40,58]. The initial estimation for parameter $k$ in each reach was determined through Equation (5) [40,58]:

$$k = L/Vw, \tag{5}$$

where $L$ is the reach length (in km) and $Vw$ is the flood wave celerity (in km.h$^{-1}$), considered as 1.5 times the average velocity [65] for the SRB, that is, 3.72 km.h$^{-1}$.

### 2.3.6. Modeling Scenarios, Calibration, Validation and Assessment of Model Performance

In this study, two types of model simulations are considered: (1) a continuous model simulation (CS) over the entire period 2006–2020 and (2) an event-based model simulation (ES) focused on major flood events selected over the period 2006–2020.

In the CS scenario, the model calibration is carried out over the period 2006–2015, while the validation is achieved over the period 2016–2020. In the ES scenario, 5 flooding events, respectively, in 2010 (flood peak: 976 m$^3$·s$^{-1}$), 2012 (flood peak: 1112 m$^3$·s$^{-1}$), 2013 (flood peak: 1074 m$^3$·s$^{-1}$), 2015 (flood peak: 893 m$^3$·s$^{-1}$) and 2020 (flood peak: 1209 m$^3$·s$^{-1}$) are selected: the first three events (2010, 2012 and 2013) are used for model calibration, while the remaining two events (2015 and 2020) are used for model validation (using the average of the calibrated parameter values on the first three events in 2010, 2012 and 2013).

In both scenarios, the optimal values for model parameters are identified during the calibration phase using the Nelder and Mead algorithm [40,58]. Additionally, the model parameters were constrained within physical admissible ranges presented in Table 1. The distributions of calibrated values across the different subwatersheds are further compared to assess significant differences between the two calibration scenarios (CS or ES) using the nonparametric Kruskal–Wallis test (at the 5% significance level).

**Table 1.** Initial estimates and ranges for the calibration of the HEC-HMS model parameters.

| Model Component | Parameter (Unit) | Initial Range/Values [1] | Maximum Range |
|---|---|---|---|
| SCS loss model | Initial abstraction ($Ia$, mm) | 9.03–16.37 | 0–500 |
| | Curve Number ($CN$, -) | 75.63–84.91 | 1–100 |
| SCS-UH transform model | Lag time ($Tlag$, min) | 1163.81–3516.57 | 3.6–30,000 |
| | Percent impervious ($Imp$, %) | 10 | 0–100 |
| Muskingum routing model | ($k$, hours) | 4.91–62.26 | 0.1–150 |
| | ($x$, -) | 0.1 | 0–0.5 |

[1] The initial range is defined by the initial values calculated across the 12 sub-watersheds in the SRB.

The model performance for both calibration and validation periods in each scenario is assessed using common guidelines for model performance evaluation [66,67], including the Nash–Sutcliffe efficiency (*NSE*, Equation (6)), the Kling–Gupta efficiency (*KGE*, Equation (7)), the coefficient of determination ($R^2$, Equation (8)), the root-mean standard deviation ratio (*RSR*, Equation (9)), the percent error in flood peak discharge (*PEP*, Equation (10)) and the percent error in flood runoff volume (*PEV*, Equation (11)).

$$NSE = 1 - \frac{\sum_i (Q_o - Q_s)_i^2}{\sum_i \left(Q_{o,i} - \overline{Q}_o\right)^2}, \tag{6}$$

$$KGE = 1 - \sqrt{(r-1)^2 + (\mu_s/\mu_0 - 1)^2 + (\sigma_s/\sigma_0 - 1)^2}, \tag{7}$$

$$R^2 = \frac{\left[\sum_i \left(Q_{o,i} - \overline{Q}_o\right) \times \left(Q_{s,i} - \overline{Q}_s\right)\right]^2}{\sum_i \left(Q_{o,i} - \overline{Q}_o\right)^2 \times \sum_i \left(Q_{s,i} - \overline{Q}_s\right)^2}, \tag{8}$$

$$RSR = \frac{\sqrt{\sum_i (Q_o - Q_s)_i^2}}{\sqrt{\sum_i \left(Q_{o,i} - \overline{Q}_o\right)^2}}, \tag{9}$$

$$PEP = \frac{Q_{s,i}^p - Q_{o,i}^p}{Q_{o,i}^p} \times 100, \tag{10}$$

$$PEV = \frac{V_{s,i} - V_{0,i}}{V_{0,i}} \times 100, \tag{11}$$

where $Q_{0,i}$ and $Q_{s,i}$ are, respectively, the observed and simulated daily discharges, $\overline{Q}_o$ is the average of observed values, $r$ is the product moment Pearson's correlation coefficient between observed and simulated values, $\mu_s/\mu_0$ is the ratio of the mean of simulated and observed discharges, $\sigma_s/\sigma_0$ the ratio of the standard deviations of simulated and observed discharges, $Q_{s,i}^p$ and $Q_{0,i}^p$ are the simulated and observed flood peak discharge for a given flood event $i$ and $V_{s,i}$ and $V_{0,i}$ are the simulated and observed flood event runoff volumes for a given event $i$. The performance metrics were further used to assess the overall model performance following the guidelines of [66,67]. The calibrated model is deemed to be good when $0.55 < R^2$, $NSE \leq 0.65$, $0.50 < KGE \leq 0.75$, $0.60 < RSR \leq 0.70$, $\pm 15\% < PEP \leq \pm 30\%$ and $\pm 15\% < PEV \leq \pm 20\%$. The model can be considered very satisfactory when $0.65 < R^2$, $NSE \leq 1.00$, $0.75 < KGE \leq 1.00$, $0.00 < RSR \leq 0.60$ and $PEP$, $PEV \leq \pm 15\%$.

2.3.7. Sensitivity Analysis

Model sensitivity analysis is helpful in assessing how the different calibrated parameters are effective within the model and to which parameters the hydrological model outputs are sensitive [68–70]. In this study, we used a single one-at-a-time (OAT) approach to assess model sensitivity, where each parameter is updated within the range of $\pm 25\%$ of its optimal value by increments of $\pm 5\%$. At each step, the relative percent change in flood peak discharge and flood runoff volume is calculated to define a sensitivity trend line, where the line slope (obtained through linear regression fit) is the sensitivity coefficient of the given parameter.

## 3. Results and Discussion
### 3.1. Calibrated Model Parameters

The distribution of the calibrated model parameters across the sub-watersheds is presented in Figure 5 and compared through boxplots for different calibration schemes. For both initial abstraction (*Ia*, Figure 5a), Curve Number (*CN*, Figure 5b), lag time (Figure 5c) and flood wave routing time (*k*, Figure 5e), there are no significant differences in the distribution of values across the sub-watersheds (Kruskal–Wallis *p*-values = 0.534–0.999), indicating that the distribution for these factors is spatially homoge-

nous and does not depend on the calibration scheme (CS or ES scenario). However, for a given parameter, substantial differences can appear between the sub-watersheds given a calibration scheme: for example, in the CS scenario, the range of calibrated values for *Ia* is 22.10–100, which could further be explained by the wide range of soil surface conditions, soil types and morphometric properties within the sub-watersheds. The calibrated values for percent impervious (Imp, Figure 5d) and flood wave attenuation factor (x, Figure 5f) show, however, significant changes in distribution between the two calibration schemes (Kruskal–Wallis *p*-values < 0.05). This indicates that these parameters are highly sensitive to the mechanisms generating flood events and should therefore be conditioned by specific antecedent conditions, resulting in specific flood runoff hydrograph shapes.

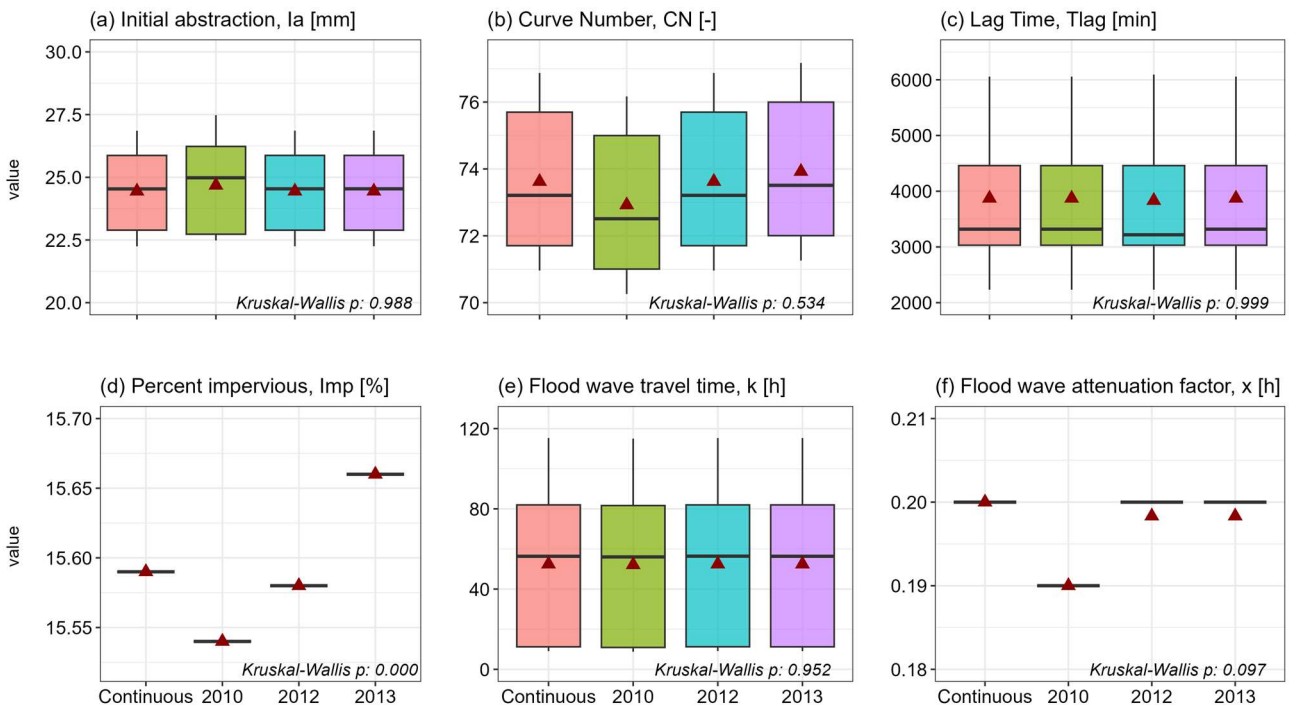

**Figure 5.** Boxplot comparison of the distribution of calibrated model parameters under CS and ES scenarios in the SRB.

### 3.2. Hydrological Model Performance

3.2.1. Continuous-Based (CS) Hydrological Simulation

In the CS scenario, carried out at the daily timestep, the model calibration is achieved over the period 2006–2015, while the validation is performed over the period 2016–2020. The results presented in Figure 6 highlight a good agreement between the simulated and observed daily discharge hydrographs over the 2006–2020 simulation period (Figure 6a). In both calibration and validation phases, the shape and trend of the simulated and observed discharge hydrographs are almost similar, with a coefficient of determination ($R^2$) of 0.87 and 0.84, respectively, in calibration and validation (Figure 6d,e). Also, the comparison of the flow duration curves shows a similar pattern in exceedance probabilities for different flood quantiles over both calibration and validation periods (Figure 6b,c), albeit with an overestimation tendency in high flows, more precisely above the 50% exceedance probability threshold. These downwards branches, highlighting the limited ability of the calibrated model to simulate low flows, could be explained by the fact that specific processes such as baseflow or groundwater flow are not explicitly considered here since the focus is surface runoff and peak flood discharge.

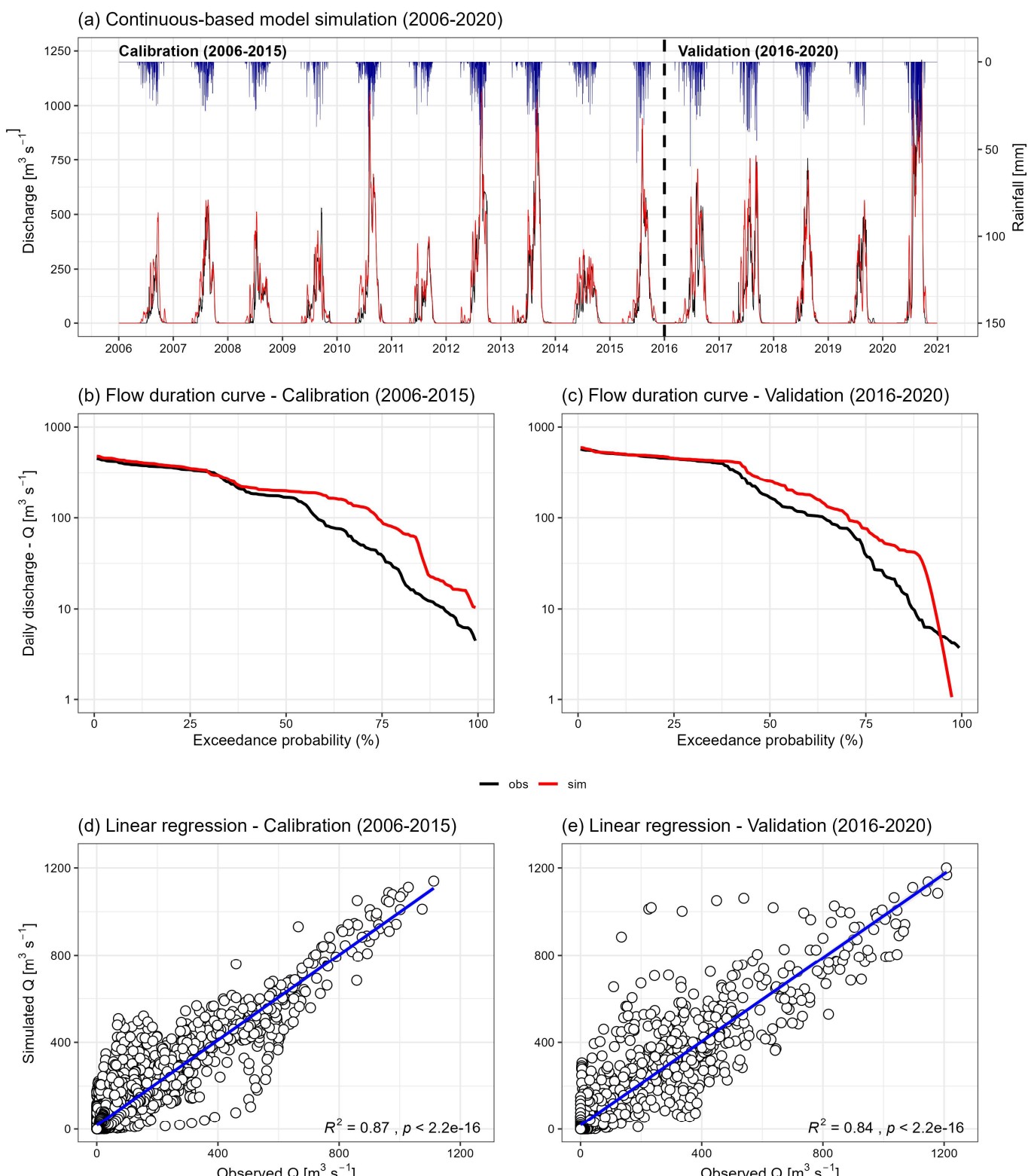

**Figure 6.** Comparison of simulated and observed flood runoff hydrographs in continuous simulation: (**a**) observed vs. simulated daily discharges over the 2006–2020 period; (**b**) simulated and observed flow duration curves in calibration period; (**c**) simulated and observed flow duration curves in validation period; (**d**) simulated and observed discharges scatterplot in calibration period; (**e**) simulated and observed discharges scatterplot in validation period.

The evaluation metrics presented in Table 2 highlight the CS performance. In the calibration period, the model tends to overestimate observed discharges. In the validation period, however, the model shifts to underestimation, although in a lesser magnitude (PEP = −0.670%). In the validation period, the simulated discharges are lower than the observed flows for the years 2018 and 2019. Overall, a relative overestimation bias of 2.48% is observed in the calibration phase, which is reduced to an underestimation of 0.67% in validation.

**Table 2.** Model performance evaluation in continuous simulation.

| Criteria | Calibration (2006–2015) | Validation (2016–2020) |
| --- | --- | --- |
| *NSE* | 0.850 | 0.813 |
| *KGE* | 0.780 | 0.820 |
| $R^2$ | 0.873 | 0.837 |
| *RSR* | 0.400 | 0.400 |
| *PEV* (%) | 21.710% | 15.220% |
| *PEP* (%) | 2.480% | −0.670% |

The remaining differences between the simulated and observed flows could be explained by the model parameter uncertainty, as discussed in [16], but also by the accuracy of the input data sources [1,3,16]; for instance, the LULC input layer used in this study for model calibration is the year 2009, which does not fully represent the soil surface conditions over the entire simulation period.

3.2.2. Event-Based (ES) Hydrological Simulation

In the case of the event-based simulation, five (05) flood events, observed between June 1st and October 31st in the years 2010, 2012, 2013, 2015 and 2020, respectively, are selected for model calibration and validation. The aim of this simulation is to assess the model's ability to simulate accurately the flood runoff hydrographs and the flood peak discharges for all events, as this generally corresponds to the maximum flood occurring downstream of the SRB in the urban city of Niamey. Also, we shall disclose that peak discharge values could appear in other months (outside the June to October period) in the SRB, but these events were ignored since they did not result in any flooding event downstream of the SRB in the urban city of Niamey.

Figure 7 compares the trend and shape between the simulated and observed runoff hydrographs, as well as the average daily cumulated runoff volumes and the linear correlations between simulated and observed discharges. A good agreement between observed and simulated flood event hydrographs is reported for the individual years 2010, 2012, 2013, 2015 and 2020 (Figure 7a,d,g,j,m), with high determination coefficients ($R^2$) of 0.98, 0.95, 0.96, 0.97 and 0.94 (Figure 7c,f,i,l,o, respectively). Also, the comparison of cumulative flood runoff volumes for these events also shows a good agreement, with marginal differences (Figure 7b,e,h,k,n).

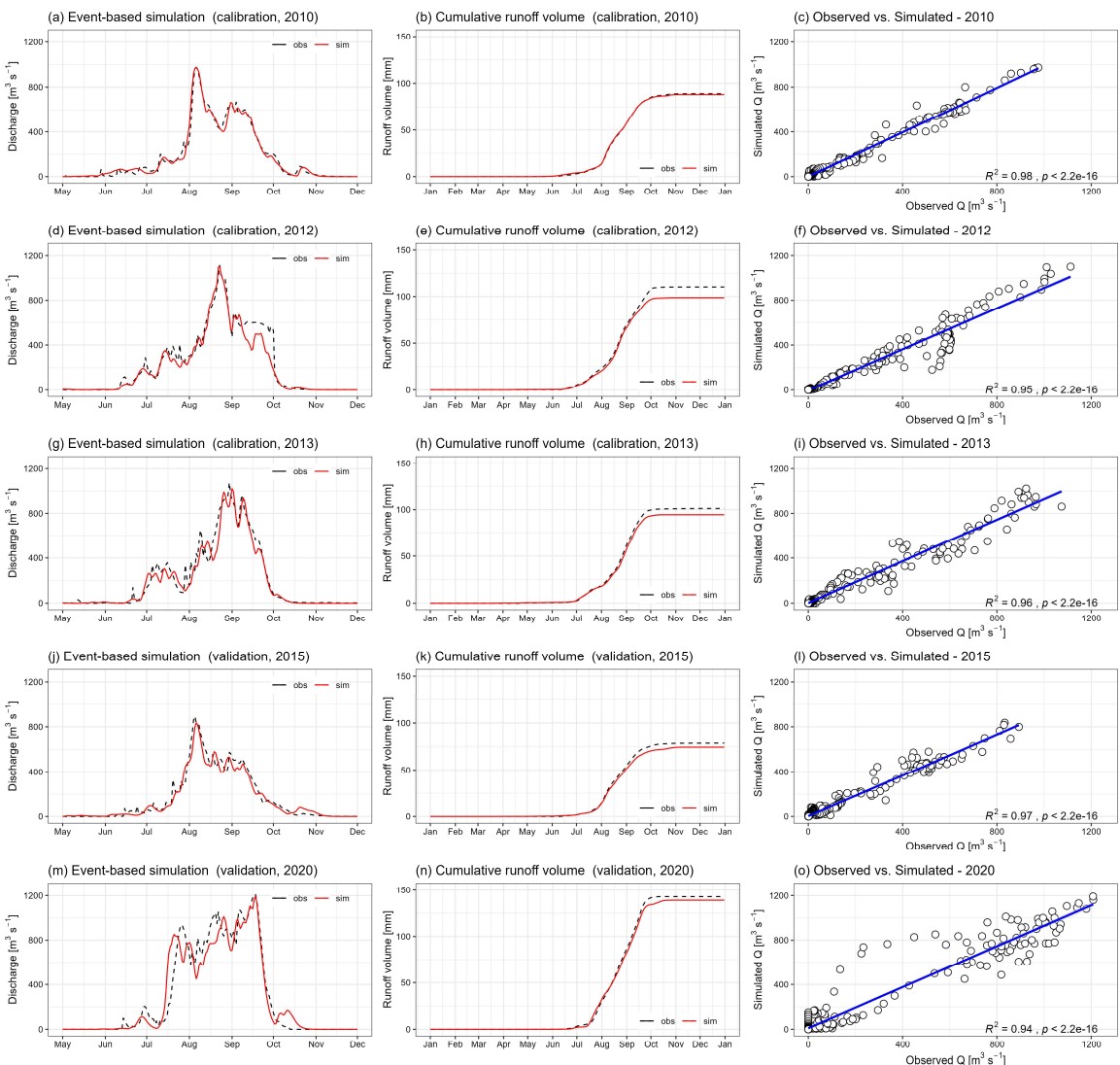

**Figure 7.** Comparison of simulated and observed flood runoff hydrographs in event-based simulation: (**a,d,g,j,m**) flood runoff hydrograph for flood events in 2010, 2012, 2013, 2015 and 2020, respectively; (**b,e,h,k,n**) cumulative runoff volume for single flood events; (**c,f,i,l,o**) observed vs. simulated scatterplots for single flood event.

Table 3 indicates that the simulated values are almost similar to the observed values for all events, therefore highlighting the satisfactory model calibration in the years 2010, 2012 and 2013. The total runoff volume error and peak flow error are, respectively, −1.13% and −0.35% for the 2010 flood event, −10.5% and −1.04% for the 2012 flood event and −6.10% and −2.10% for the 2013 flood event. For the calibration period, the values of the NSE, KGE and RSR coefficients are 0.96, 0.92 and 0.17, respectively, indicating a very satisfactory model performance, which also outperforms the calibration metrics obtained in the CS scenario.

It should be noted that a 2-day lag was observed in 2013 between the simulated and observed flood runoff hydrographs, as well as an underestimation of peak discharge in almost all events. The lag between the onset of the simulated and observed peak discharge could be due to the uncertainty around the calibrated flow travel time and the initial loss values [71,72]. Such discrepancy has been reported in earlier applications of the HEC-HMS model: for instance, refs. [65,73] reported a 1–3 h shift between observed and simulated peak flow times in almost all events in a basin model developed in India, while [74] also observed 15–96 min shifts in peak flow in a semi-arid river basin in northern China.

**Table 3.** Model performance evaluation in event-based simulation.

| Criteria | Calibration (2010 Event) | Calibration (2012 Event) | Calibration (2013 Event) | Validation (2015 Event) | Validation (2020 Event) |
|---|---|---|---|---|---|
| $NSE$ | 0.98 | 0.95 | 0.96 | 0.96 | 0.94 |
| $KGE$ | 0.98 | 0.87 | 0.91 | 0.90 | 0.93 |
| $R^2$ | 0.98 | 0.95 | 0.96 | 0.97 | 0.94 |
| $RSR$ | 0.10 | 0.20 | 0.20 | 0.20 | 0.30 |
| Observed peak discharge ($m^3/s$) | 975.6 | 1112.3 | 1074.2 | 893 | 1208.6 |
| Simulated peak discharge ($m^3/s$) | 972.2 | 1100.7 | 1021.3 | 835.7 | 1191.2 |
| $PEP$ (%) | −0.35% | −1.04% | −4.92% | −6.42% | −1.44% |
| Observed timing of peak [1] | 08/06 | 08/23 | 08/30 | 08/05 | 09/17 |
| Simulated timing of peak [1] | 08/06 | 08/23 | 09/01 | 08/06 | 09/18 |
| Observed volume (mm) | 89.81 | 111.24 | 102.08 | 79.48 | 143.85 |
| Simulated volume (mm) | 88.79 | 99.54 | 95.28 | 75.09 | 140.03 |
| $PEV$ (%) | −1.13% | −10.50% | −6.66% | −5.52% | −2.65% |

[1] The observed and simulated dates are provided in the MM/DD format.

The flood events in 2015 and 2020 were used for model validation in this study. It should also be noted that the magnitude of these flood events is unprecedented at the Garbey Kourou hydrometric station. The model validation shown in Figure 7 clearly highlights a good agreement between the observed and simulated flow runoff hydrographs. However, a 1-day lag in the timing of the peak discharge is observed between the measured and simulated flow, which could again be attributed to the flow travel/routing time in the river basin [65,73].

*3.3. Sensitivity Analysis*

The sensitivity analysis is carried out in this study based on OAT relative changes in optimal model calibrated values by small increments of 5% within the ±25% relative change range. The relative change in flood peak discharge and flood runoff volume are selected as targeted responses to assess their sensitivities to the model parameters presented in Figure 8. These results indicate that *CN*, *Tlag* and *k* are the most sensitive parameters for peak flood discharge (Figure 8a), with sensitivity coefficients of 0.53, −0.23 and −0.22, respectively. *CN* is the most sensitive parameter for flood runoff volume (Figure 8b) with a sensitivity coefficient of 0.41. The least sensitive parameters overall are Imp and x for both responses, with respective sensitivity coefficients of 0.01 and −0.02 for peak flood discharge and 0.04 and 0.00 for flood runoff volume.

Also, it should be noted that while *Tlag* and *k* appear to be significantly affecting peak flow discharges (Figure 8a), they are marginally affecting flood runoff volumes (Figure 8b). Also, positive sensitivity coefficients, such as those of *CN*, suggest that an increase in the corresponding parameter translates as an increase in peak discharge and flood runoff volume; meanwhile, negative sensitivity coefficients, such as those of *Tlag* and *k*, suggest that an increase in these parameters translates as a decrease in flood peak discharges. These results also shed light on how the different model parameters control surface runoff generation, as represented within the model, but also provide useful insights for the assessment of simulated discharge uncertainties since the highly sensitive parameters should be estimated with a narrow error to minimize the overall model uncertainty.

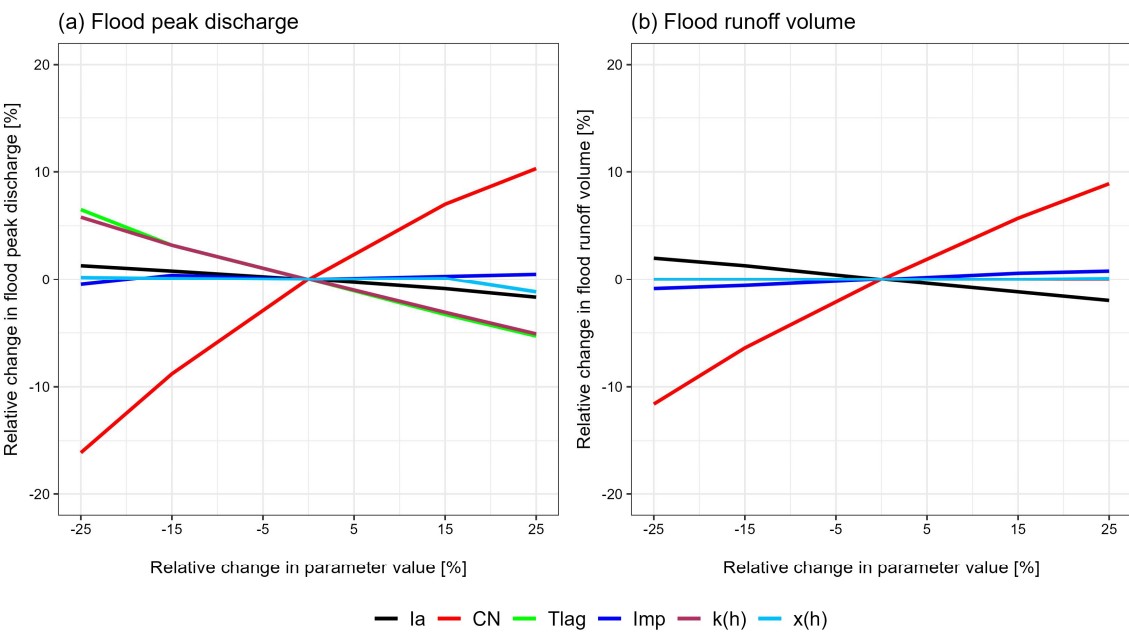

**Figure 8.** HEC-HMS model sensitivity analysis in the SRB. (**a**) Sensitivity of flood peak discharge to model parameters. (**b**) Sensitivity of flood runoff volume to model parameters.

## 4. Discussion

This study focused on the application of the HEC-HMS hydrological model in the Sirba River Basin (SRB), a transboundary area shared between Burkina Faso and Niger in West Africa. In the previous literature, several modeling applications have already been conducted in the SRB for a wide range of applications. The ABC hydrologic model has been used to assess the influence of future climate on discharges in the SRB [75]. Ref. [23] used the *Soil and Water Integrated Model* (SWIM) for land use and climate change attribution changes on river flooding. The *Soil and Water Assessment Tool* (SWAT) was also applied in the SRB to analyze changes in flow regimes driven by climate variability [34]. Refs. [31,32] used multiple linear regression equations based on sea-surface temperature predictors for statistical seasonal rainfall–streamflow modeling and forecasting in the SRB. Ref. [35] analyzed how process refinement could improve hydrological modeling with the *Hydrological Predictions for the Environment* (HYPE) model. Refs. [36,38] used the lumped and catchment scale *Identification of Unit Hydrographs and Component Flows from Rainfall, Evaporation and Streamflow* (IHACRES) model for rainfall–runoff modeling and the assessment of hydroclimatic uncertainties under climate change. Ref. [39] used the *Global Flood Awareness System* (GloFAS) model to develop an early warning system in the SRB, highlighting the practical use of hydrological modeling in developing risk mitigation strategies. Ref. [37] applied the same model to develop a regional hydrological forecast for operational early warning. Ref. [30] applied the *Variable Infiltration Capacity* (VIC) model over the 1981–2010 period in the SRB. Overall, these hydrological modeling applications in the SRB have contributed to the development of early warning systems, the improvement of flood mitigation strategies and the assessment and quantification of the major hydrological processes. They also underscore the practical relevance of hydrological modeling in addressing the challenges posed by floods and water security, especially for the region.

The HEC-HMS model calibration and parameter sensitivity analysis were carried out over the period 2006–2020 in this study. The range of values for calibrated model parameters is essential for ensuring the accuracy of the simulated processes and model outputs. It should be noted that potential evapotranspiration was not considered among the model inputs, which was not a limiting factor to attaining satisfactory model performance. This could further be explained by the fact that most hydrological models are

rarely sensitive to potential evapotranspiration [4,76,77], which typically affects antecedent soil moisture levels between rainfall events, especially in the case of continuous-based simulations of the water cycle. Also, while intricate relationships in neighboring contexts between surface runoff and groundwater have been reported, such as preferential recharge pathways or groundwater feedback through rising tables [78,79], no such significant control from groundwater to surface runoff was observed in this study. In fact, previous studies highlighted that in the SRB, flood magnitude is largely influenced by surface processes rather than groundwater flow, as the river basin hydrology remains closely dependent on rainfall variability [33].

This study focused on the comparison of two types of simulations, i.e., continuous vs. event-based simulation. The results indicated marginal differences in the spatial distribution of the calibrated model parameters, only substantial for parameters such as the percent impervious and the flood wave attenuation factor. In general, continuous-based simulation provides a comprehensive understanding of long-term runoff behavior, which is essential for water resource management and planning, while event-based simulation is valuable for analyzing specific flood events and their impacts. However, continuous-based simulation has been noted to have disadvantages, such as inaccurate extrapolations, extensive data requirements and the potential loss of sharp events when the modeling timescale is large [80,81]. Such types of events are generally well captured in the case of event-based simulations, providing a more detailed understanding of the hydrological response to these events and their associated impacts.

Regarding sensitivity analysis, this study used a one-at-a-time approach to evaluate the sensitivity of the calibrated model to each parameter through the evaluation of the relative change in specific model responses (flood peak and volume). The most sensitive parameters reported in this study are the Curve Number, lag time and the flood wave travel time factors for flood peak, while only the Curve Number parameter was found to be sensitive for flood runoff volume. These findings could be compared to other studies that reported the percent impervious followed by the groundwater coefficient to be the most sensitive parameters [82,83]. Ref. [84] identified that for peak volume, their model was found to be most sensitive to the loss parameters and the storage coefficient. Sensitivity analysis studies in the Upper Indus River Basin suggested that the most important parameters were storage parameters [85]. These findings highlight the variability of sensitivity analysis results across different hydrological contexts, likely affected by rainfall–runoff dynamics, climate and land surface interaction and soil surface conditions.

## 5. Conclusions

This study focuses on the simulation of the rainfall–runoff relationship in the Sirba River Basin (SRB), which is a major tributary of the Niger River. The HEC-HMS model is used with two simulation schemes, i.e., a continuous and an event-based model simulation, over the period 2006–2020. The continuous-based simulation used the subperiod 2006–2015 for calibration and 2016–2020 for validation. For the event-based model, the calibration and validation relied on five significant flood events that occurred in 2010, 2012, 2013 (for model calibration), 2015 and 2020 (for model validation). Overall, the results demonstrated that the HEC-HMS model successfully simulates surface runoff and daily discharges in the Sirba watershed, with satisfactory accuracy for flood peak and flood runoff volumes. The study also reported that while a satisfactory performance is always attained under both continuous and event-based simulations, some parameters calibrated values are affected by the modeling scheme, therefore suggesting that the choice of such calibration should be critically selected depending on the desired model application. It was also noted that the event-based simulation performed better overall than the continuous-based simulation for both flood peak discharge and flood runoff volume based on the performance metrics used to assess model performance. The model sensitivity analysis was also performed to identify the most influential parameter in the system. The results showed that the Curve Number, lag time and flood wave travel time were the most sensitive model parameters,

while initial abstraction, flood wave attenuation factor and percent impervious are the least sensitive parameters in the context of the SRB. The study concludes that the selected methods to simulate the hydrological processes are effective for similar watersheds (in terms of topography, hydrology, land use and soil types) to that of the SRB, therefore allowing for the development of appropriate flood control policies and infrastructure.

**Author Contributions:** Conceptualization, I.S.T., R.Y., D.N., M.M.A. and H.K.; methodology, I.S.T., R.Y., D.N., A.K., M.M.A. and H.K.; software, I.S.T. and R.Y.; validation, I.S.T., R.Y., D.N., A.K., M.M.A. and H.K.; formal analysis, I.S.T., R.Y. and D.N.; investigation, I.S.T., R.Y. and D.N.; resources, D.N., M.M.A. and H.K.; data curation, I.S.T. and R.Y.; writing—original draft preparation, I.S.T. and R.Y.; writing—review and editing, I.S.T., R.Y. and D.N.; visualization, I.S.T. and R.Y.; supervision, D.N., A.K., M.M.A. and H.K.; project administration, D.N., A.K., M.M.A. and H.K.; funding acquisition, D.N., M.M.A. and H.K. All authors have read and agreed to the published version of the manuscript.

**Funding:** This research was funded by the World Bank through the African Center of Excellence Impact (ACE-Impact) Program (Grant Number: IDA 6388-BF/D443-BF).

**Data Availability Statement:** The input climate data used in this study can be obtained upon request to the ANAM-BF and DMN-Niger. The code, data and figures produced in this study can be shared upon reasonable request to the corresponding author.

**Acknowledgments:** The authors are grateful to the National Meteorology Agency in Burkina Faso (ANAM-BF) and the National Meteorological Directorate of Niger (DMN-Niger) who provided the rainfall and discharge data used in this study. They are also grateful to the World Bank through the African Center of Excellence Impact (ACE-Impact) Program for the financial support.

**Conflicts of Interest:** The authors declare no conflicts of interest.

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
