# Peer review of "Daily Simulation of the Rainfall–Runoff Relationship in the Sirba River Basin in West Africa: Insights from the HEC-HMS Model"

_hydrology, doi:10.3390/hydrology11030034_

Round 1
Reviewer 1 Report
Comments and Suggestions for Authors
The paper is of a fairly good standard and disproves the common impression that the CN method is not suitable for large catchments. Of course, the above statement applies mostly to the hourly step. I have a few questions about the article that need clarification.
1) In the continuous-based model, zero runoff values are obtained in Figure 6a. In the case of the continuous-based model, this is a somewhat unusual result, assuming that the river does not actually dry up (rainy and dry periods). Is this also a case of complete drying of the flow? Alternatively, what methods could be used to solve the problem?
2) It is generally known that the CN method is very convenient to use single episodic rainfall (does not consider drying up of the catchment)? How have changes in initial loss (or CN) over relatively long rainfall episodes been calculated for a continuous-based model?
3) Were individual parameters varied during the event-based modelling, or were they set at the beginning and then left in the no-impact mode?
4) The paper lacks an evaluation of peak flows (continuous-based model). In the case of floods, the ability to correctly predict (simulate) peak flow is critical (building mobile flood control measures, evacuating areas). Assuming the simulated profile is not directly above the reservoir. In the case of event-based models, part of the results can be read from Figure 7.
In general, I evaluate the paper positively. Within the paper, analyses have been made regarding the influence of the different parameters as well as comparisons between episodic and continuous simulations.
Remark:
In the case of the results for Figures 6d and 6e, downward branches are also shown, which in general, without considering groundwater runoff, rather degrade the results and are not interesting unless we consider the simulation for reservoirs.
Author Response
Dear Reviewer 1,
Please find attached our responses to your valuable comments.

Reviewer 2 Report
Comments and Suggestions for Authors
(1)In the discussion section, there are a lot of studies in the SRB concerning hydrologic modelling, could you please give more introduction for it in the section Introduction?
(2)Concerning the data used in the calibration and validation ,why doesn't use the years 2021,2022 for modelling ?
(3)due to the middle-size basin ,daily simulation of the rainfall-runoff relationship is enough to do flood process prediction and warning analysis?
(4) in the figure 5,6, there are the problems missing titles of abscissa or ordinate
(5)in the line 291-292, only LULC 2019 is consided for simulation ,what is the influence for simulation?
Comments on the Quality of English Language(1)In the discussion section, there are a lot of studies in the SRB concerning hydrologic modelling, could you please give more introduction for it in the section Introduction?
(2)Concerning the data used in the calibration and validation ,why doesn't use the years 2021,2022 for modelling ?
(3)due to the middle-size basin ,daily simulation of the rainfall-runoff relationship is enough to do flood process prediction and warning analysis?
(4) in the figure 5,6, there are the problems missing titles of abscissa or ordinate
(5)in the line 291-292, only LULC 2019 is consided for simulation ,what is the influence for simulation?
Author Response
Dear Reviewer 2,
Please find attached our responses to your valuable comments.

Reviewer 3 Report
Comments and Suggestions for Authors
Firstly, I would like to congratulate the authors on their work. As a suggestion, it would be important to highlight the difference between this study and other research already conducted, emphasizing its unique contributions. Additionally, it would be interesting to compare the predictive performance with other methods, possibly including a naive model. Another performance metric that should be considered is RMSE. Looking ahead, potential perspectives could include issues such as generating predictions through ensemble methods and incorporating artificial intelligence.
Comments on the Quality of English LanguageEnglish with easy understanding. Perhaps it would be interesting to write short sentences.
Author Response
Dear Reviewer 3,
Please find attached our responses to your valuable comments.

Round 2
Reviewer 1 Report
Comments and Suggestions for Authors
From my point of view, I think the authors have incorporated my comments, and I recommend that the article be accepted. The modelling could be extended to work with various scenarios and compare with other methods. However, this would be more appropriate for a separate article.